# Determination of Electron Beam Energy in Measuring the Electron-Impact Ionization Cross Section of He-like $Fe^{24+}$

Yang Yang [1],*, Dipti [2],†, Amy Gall [3], Galen O'Neil [4], Paul Szypryt [4], Adam Hosier [1], Adam Foster [3], Aung Naing [2], Joseph N. Tan [5], David R. Schultz [6], Randall Smith [3], Nancy Brickhouse [3], Yuri Ralchenko [5] and Endre Takacs [1,2]

[1]  Department of Physics and Astronomy, Clemson University, Clemson, SC 29634, USA
[2]  Associate, National Institute of Standards and Technology, Gaithersburg, MD 20899, USA
[3]  Center for Astrophysics | Harvard & Smithsonian, Cambridge, MA 02138, USA
[4]  National Institute of Standards and Technology, Boulder, CO 80305, USA
[5]  National Institute of Standards and Technology, Gaithersburg, MD 20899, USA
[6]  Department of Astronomy and Planetary Science, Northern Arizona University, Flagstaff, AZ 86011, USA
*  Correspondence: yy4@g.clemson.edu
†  Present address: International Atomic Energy Agency, A-1400 Vienna, Austria.

**Abstract:** In an effort to measure electron-impact ionization (EII) cross-sections of He-like $Fe^{24+}$ at the electron beam ion trap (EBIT) facility of the National Institute of Standards and Technology (NIST), we have experimentally determined the corrections to the nominal beam energy determined by the voltages applied to the EBIT. High-resolution X-ray spectra were recorded at nominal electron beam energies between 6660 eV and 6750 eV using X-ray microcalorimetry based upon an array of 192 transition-edge sensors (TES). A large-scale collisional-radiative simulation of the non-Maxwellian EBIT plasma using relevant atomic data calculated with Flexible Atomic Code allowed us to determine the space-charge correction due to the electron beam including the neutralization factor by the ion cloud of the EBIT.

**Keywords:** EBIT; beam energy; ionization cross-section; space-charge correction; neutralization

## 1. Introduction

Accurate atomic data of the electron-impact ionization cross-sections are important for modeling high temperature plasma and benchmarking state-of-the-art theoretical calculations. The electron-impact ionization cross-sections of different charge states of iron are particularly important, since iron is a highly abundant element in the universe, and these data are critical in the interpretation of astrophysical observations [1]. For example, Hahn et al. [2] pointed out the importance of the measured experimental ionization cross-section of $Fe^{17+}$ on the ionization rate coefficients that determine the ionization balance in astrophysical plasmas. In turn, the ionization balance is a critical factor that affects the emission intensities of diagnostic spectral lines in astrophysical observations.

Taking advantage of the experimental capabilities of the electron beam ion trap (EBIT) at the National Institute of Standards and Technology (NIST) [3], the electron-impact ionization cross-section of He-like $Fe^{24+}$ can be measured at different electron beam energies. There are, however, various experimental factors that need to be taken into account to assess the different sources of uncertainty of the measured experimental data in order to make a proper comparison with advanced theoretical calculations. The most significant experimental factors include the density of the electron beam, the overlap between the electron beam and the ion cloud, the density of neutral atoms in the trap, the efficiency curve of the spectrometer components and the detector, as well as the experimental accuracy in determining the energy of the electron beam.

The latter is especially important at energies near the ionization threshold, as the ionization cross-section changes rapidly in this region [4–7]. In the EBIT, the electron beam energy is primarily determined by the difference between the potential on the central trap electrode (drift tube) and that on the electron gun cathode, but it is also affected by the space-charge of the electron beam and the neutralization of the electrons by the trapped ions. In this paper, we present a detailed analysis to determine the total space-charge correction that includes both of these by analyzing the intensity changes of dielectronic recombination (DR) satellite lines near the He-like resonance line of $Fe^{24+}$. The resonance nature of the DR process allows the measurement of the actual electron energies that differ from the nominal beam energies by the space-charge corrections, including the shielding of the electron beam by the ion cloud characterized by the so-called neutralization factor. The details of the measurement and the ionization cross-sections themselves will be presented in an upcoming publication.

## 2. Experimental Details

The measurements were taken at the NIST EBIT facility. A detailed description of the vertically oriented versatile electron-beam-excited radiation source can be found elsewhere [3]. In short, the NIST EBIT is a cylindrically symmetric system used to produce and trap highly charged ions. It has three main components: the electron gun, the drift tube structure (ion trap), and the collector assembly. In the EBIT, a quasi-mono-energetic, intense electron beam is emitted from a curved-surface barium-doped cathode in the electron gun assembly.

The electron beam is compressed to approximately 35 μm radius ($\sim 10^{11}$ electrons/cm$^3$) by a superconducting magnet capable of producing an up to 2.7 T magnetic field in the trap section of the machine. The electron beam interacts with the ion cloud in the trap region that consists of three cylindrical drift tube electrodes. They each have a 500 V power supply that electrically floats on top of the high voltage of the shield electrode, which is capable of floating up to 30 kV voltage, to create the trap. During the measurements, singly charged ions were injected into the trapping region along the vertical axis from a metal vapor vacuum arc (MeVVA) ion source [8]. Trapped ions are further ionized to high charge states by successive electron-impact ionization.

The relative voltages placed on the three drift tube electrodes were set to trap the ions electrostatically in the axial direction. For this particular experiment, the lower drift tube was set to +500 V and the upper drift tube was set to +250 V higher than the middle drift tube voltage. The potential difference between the cathode of the electron gun and the middle drift tube (MDT) in the trap defines the nominal electron beam energy that is modified by the space-charge of the electron beam [9,10] described in the next section. The electron beam current in these measurements was set to 100 mA.

Our current measurements focus on the determination of the total space-charge correction to the nominal electron beam energy. This includes the determination of the neutralization factor (mathematically defined in the next section), which is the space-charge correction due to the ion cloud with respect to that of the electron beam alone. Measurements were made at nominal beam energies of 6660 eV, 6710 eV, and 6750 eV. At these energies, prominent DR features are present in the spectra that allow the determination of the space-charge correction to the nominal electron beam energy set by the voltages applied to the EBIT electrodes.

DR is a two-step resonance process where a continuum electron recombines with an ion while a bound atomic electron is excited. The radiative de-excitation of the doubly-excited state produces the spectral footprints of the resonant excitation that can be observed in the emitted spectra. The specific kinetic energy requirement for the captured free electron, $E_{1e} + E_{1b} = E_2$ (see Figure 1 for the notation), makes DR produced satellite lines useful for probing the energy of the electron beam of the EBIT. It practically provides two features useful for measuring the space-charge correction: (1) high sensitivity to electron beam energy and (2) low uncertainty on input parameters to theoretical calculations. At NIST,

the energy resolution of the recently installed transition-edge-sensor (TES) spectrometer allows the observation of the DR spectral features with high X-ray energy resolution that can be matched to theoretical models.

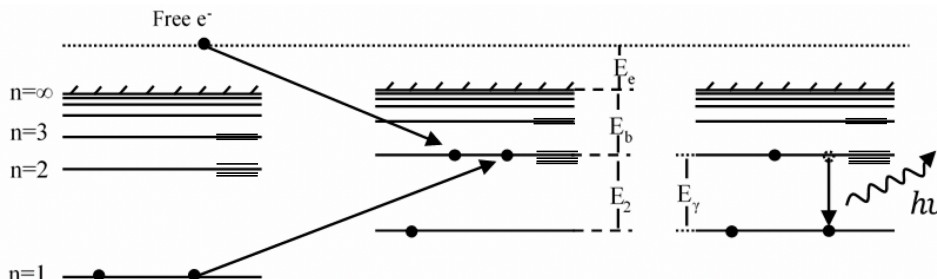

**Figure 1.** The DR process has a resonant nature, which arises from energy matching between the incoming free electron ($E_{1e}$) and the binding energy of the level it is captured to ($E_{1b}$) with the excitation energy ($E_2$) of the second electron [11]. Reproduced with permission from Gall, Amy Christina, Inner Shell Atomic Processes in Highly Charged Argon EBIT Plasma Relevant to Astrophysics; published by Clemson University TigerPrints All Dissertations, 2019.

The spectra emitted by the cloud of iron ions of the NIST EBIT were recorded with an X-ray microcalorimeter based upon an array of 192 TES detectors [12]. The energy resolution of the recently installed X-ray spectrometer allows the observation of these spectral features that can be matched to accurate theoretical models of the emission spectra. The instrument covers the 500 eV to 8 keV X-ray energy range with an energy resolution better than 5 eV over the entire region. Calibration of the instrument was performed using K$\alpha$ emission lines of Mg, Al, Fe, Co, and Ni produced by an external fluorescent source [12].

## 3. Space-Charge Correction

With the electron gun cathode grounded, the nominal electron beam energy ($E$) in the trap is determined by the potential ($V$) applied to the middle drift tube according to the expression $E = q_e V$, where $q_e$ is the absolute value of the charge of an electron. The presence of the electron beam itself, however, modifies the electric potential in the center of the trap due to the associated image charges that appear on the trap electrodes. This image-charge potential, also called space-charge potential, is negative and will lower the overall electron beam energy [13,14].

To estimate the upper limit of the space-charge offset of the electron beam energy, we can assume that the electron beam is a long cylinder with a charge per unit length of $\lambda = \frac{I}{v}$, and the number of ions in the trap are negligible. Here, $I$ is the electron beam current and $v$ is the electron velocity that to the first order can be derived from the nominal kinetic energy of the electrons. In this arrangement, we can calculate the radial potential $V_{sc}^e(r)$ outside the electron beam using Gauss's law on electrostatics [10].

$$V_{sc}^e(r) = \frac{\lambda}{2\pi\epsilon_0} \ln \frac{r}{r_e} + V_0^e \tag{1}$$

In this expression, $r_e$ is the electron beam radius, $\epsilon_0$ is the free space permittivity, $r$ is the distance from the center of the electron beam, and $V_0^e$ represents the potential at the radius of the electron beam relative to the center of the beam. For a uniform cylindrical electron density distribution, $V_0^e = \frac{\lambda}{4\pi\epsilon_0}$; and for a Gaussian distribution of the same width and total linear charge density, $V_0^e = 1.08 \frac{\lambda}{4\pi\epsilon_0}$, as was shown by [10].

The radial density distribution of the electron beam in the EBIT has been measured previously (see, e.g., [15,16]), and it is close to Gaussian in nature. An average value, equal to the density of a uniform cylindrical beam of a certain radius, can be used to characterize the electron beam density as it is producing a radial potential that is similar to

that of a Gaussian beam [10,17]. In this work, we assumed that the electron beam density distribution is Gaussian, and $V_0^e$ was calculated accordingly.

With this, the radial potential due to the charge of the electron beam can be estimated as

$$V_{sc}^e(r) = \begin{cases} (2\ln\frac{r}{r_e} + 1)V_0^e & \text{if } r > r_e \\ (\frac{r}{r_e})^2 V_0^e & \text{if } r \leq r_e \end{cases} \tag{2}$$

Setting the boundary condition for the potential at the wall of the MDT to be determined by the nominal voltage applied, the modified potential at the center of the trap needs to be corrected by Equation (2). The thesis of Gall provides examples and tables for the space-charge correction at different beam energies and beam currents in the NIST EBIT [11].

One of the missing elements of the above space-charge correction calculation is the presence of the ions in the trap that partially neutralize the charge of the electron beam. This can be characterized by the neutralization factor defined by

$$N = V_{sc}(R)/V_{sc}^e(R) \tag{3}$$

where $R$ is the radius of the MDT, $V_{sc}^e$ is the space-charge potential due to the electron beam (from Equation (2)), and $V_{sc}$ is the space-charge potential due to the electron beam shielded by the ion cloud. If the ions would completely shield the electron beam, this factor would be 0. If no ions were present, the factor is 1. The neutralization factor is generally unknown; therefore, $V_{sc}^e$ is an upper limit to the total space-charge correction $V_{sc} = N \times V_{sc}^e$.

The total space-charge correction, including neutralization by the ions, can be experimentally determined by observing processes that require specific electron energies to take place. An example is the comparison of theoretical resonance electron energies of DR satellite features with measured values, which is the approach we have followed in this work. To this end, we collected experimental spectra at three different nominal beam energies of 6660 eV, 6710 eV, and 6750 eV, where we expected that DR resonance satellite features can be observed in the spectra. The lowest nominal beam energy 6660 eV is, already without any space-charge corrections, below the 6700.4 eV [18] excitation threshold of the resonance line ($1s2p\ ^1P_1 - 1s^2\ ^1S_0$) in He-like $Fe^{24+}$; therefore, only DR satellites lines can contribute to the intensity of the feature near this energy (see Figure 1). The other two energies fall in the beam region where the space charge correction can potentially lower the actual beam energy to below the direct excitation thresholds of features observed in this X-ray regime. Whether this takes place or not depends on the neutralization factor by the ion cloud.

To interpret the observed features, collisional-radiative simulations were performed with the well-established NOMAD code for non-Maxwellian plasmas [19] that included bare and H-like to Be-like Fe ions. All the required atomic data, such as energies, transition probabilities, autoionization rates, and collisional cross-sections were calculated using the Flexible Atomic Code [20]. Rates of electron-collision processes were determined assuming a Gaussian electron energy distribution function with full width at half maximum (FWHM) of 40 eV, representing the energy spread due to the experimental electron beam profile [21]. The model included atomic configurations with single-electron excitations, with principal quantum numbers $n$ up to 15, as well as doubly-excited configurations. More precisely, singly-excited configurations $1s^2nl$ ($n \leq 15$), and autoionizing states with K-shell electron excitation, i.e., $1s2lnl'$ ($n \leq 15$) and $1s3lnl'$ ($n \leq 6$), have been considered for Li-like Fe.

Figures 2–4 show the comparison of the normalized experimental and theoretically simulated spectra at nominal beam energies of 6660 eV, 6710 eV, and 6750 eV. Spectra were normalized to the strong feature in the spectra at around 6700 eV, minimizing the least-squares difference between the intensities of the experimental and theoretical data provided the best beam energy matches for the experimental spectra. At 6660 eV and 6710 eV nominal beam energies, this resulted in satisfactory agreements (Figures 2 and 3); however, for the spectrum at 6750 eV, no good agreement was found within the framework of the current theoretical model parameters.

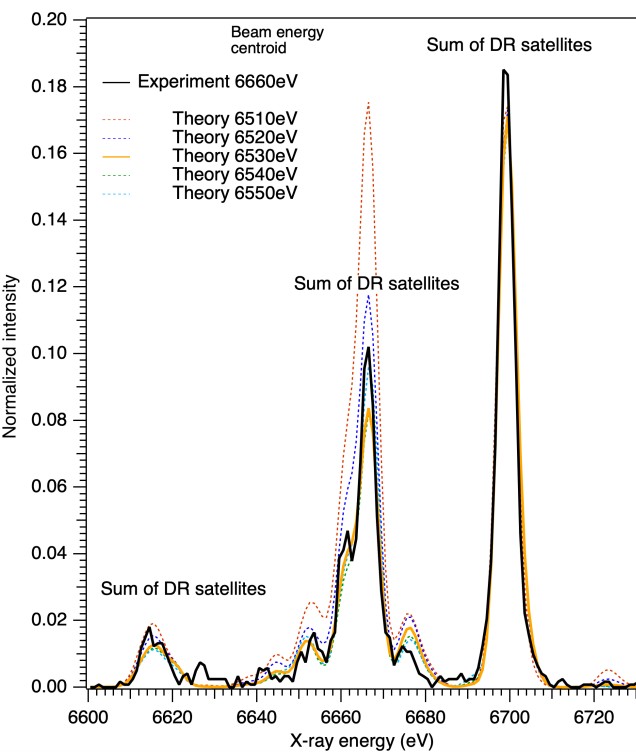

**Figure 2.** Comparison of normalized measured and theoretical spectra of Fe ions at the nominal experimental beam energy of 6660 eV. The observed spectrum is shown by a solid black line and the best theoretical match is shown by a solid orange line.

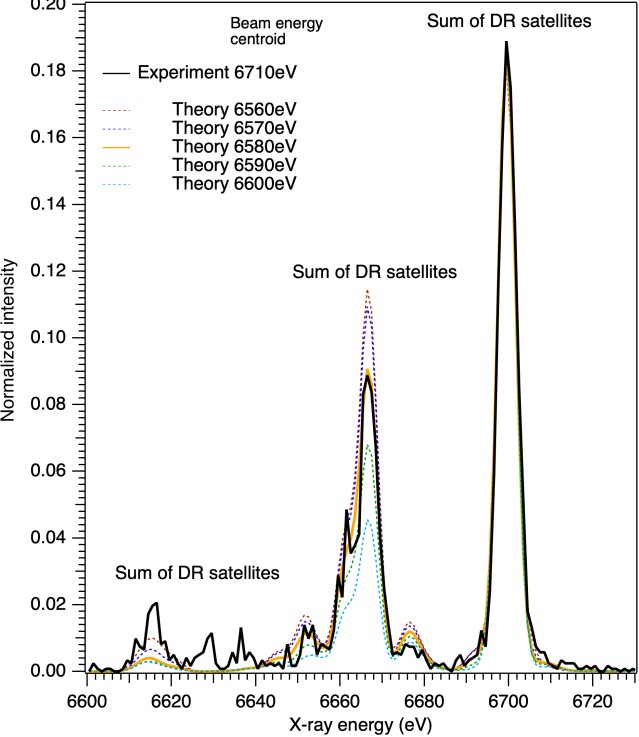

**Figure 3.** The same as Figure 1 at the nominal experimental beam energy of 6710 eV.

The DR features in these spectra in Figures 2 and 3 are mainly from Li-like Fe, and the strongest lines correspond to the $1s2lnl'–1s^2nl'$ ($n = 8$) transitions. In addition, we also observed the DR emissions from $n = 7$ and $n = 9$. From the best matches between the experiment and theory, the space-charge correction potentials in the center of the trap were

found to be 130 eV at both nominal beam energies (6660 eV and 6710 eV). We estimate the uncertainty of these determinations to be better than 10 eV based on the variation of the theoretical spectra with the beam energy assumed.

The 6750 eV nominal beam energy experimental spectrum in Figure 4 presents an interesting scenario for the theoretical model. Applying the same 130 eV space-charge correction to the nominal beam energy as determined from the other two cases, and considering the energy spread of the electron beam, the feature at around 6636 eV photon energy comes from both the direct excitation of the He-like $1s2s\ ^3S_1 - 1s^2\ ^1S_0$ transition and Li-like satellite lines due to high-$n$ DR excitations. Since the direct excitation cross-section near the excitation threshold has a large theoretical uncertainty, proper modeling of the DR satellites is not sufficient to determine the space charge correction at this nominal beam energy. Further investigation of this region, however, is interesting considering the astrophysical consequences of the different contributions. Refining the electron impact excitation cross-section near this threshold region will most likely bring closer agreement between experiment and theory.

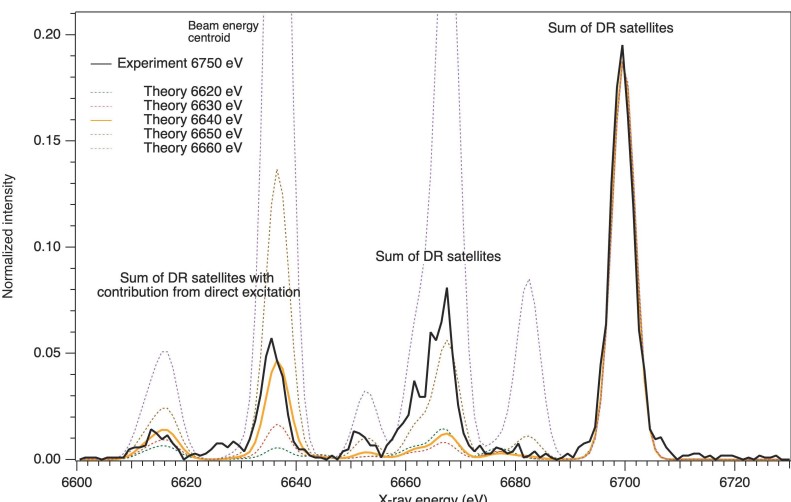

**Figure 4.** Comparison of normalized measured and theoretical spectra of Fe ions at a nominal experimental beam energy of 6750 eV. The observed spectrum is shown by a solid black line and the theoretical spectrum that corresponds to the best match is shown by a solid orange line.

## 4. Results and Conclusions

Comparison of the space-charge correction from the 6660 eV and 6710 eV nominal energy experimental spectra with Equation (2) allowed us to determine that the average neutralization factor by the ion cloud, defined by Equation (3), is 60% $\pm$ 5%. The uncertainty of the value corresponds to the upper limit of the uncertainty of the space-charge correction to the nominal beam energies. The latter was determined by matching the experimental spectra with a theoretical model of the spectral features.

These measurements of the electron beam neutralization factor will aid our experimental effort to determine the ionization cross-section of He-like Fe ions at several electron beam energies. Accurate experimental data of electron-impact ionization cross-sections are essential for the interpretation of astrophysical observations and for determining the precision of diagnostic methods of astrophysical plasma. The determination of the total space-charge correction, including neutralization in the EBIT, is crucial for the accuracy of the experimental cross-sections, especially at electron energies near the ionization threshold.

The neutralization factor determined this way relies on the accurate collisional-radiative modeling of the experimental spectra that include DR satellites to He-like spectral lines. These calculations have been tested in many prior experiments in the EUV and X-ray spectral regions. Our future efforts are going to focus on the scaling of the neutralization corrected space-charge shift to beam energies where the ionization cross-section determina-

tions will take place. Our measurements provide benchmarks for models of the ion cloud distribution in electron beam-based devices such as the EBIT.

**Author Contributions:** Conceptualization, Y.Y., A.G., E.T., D. and Y.R., Investigation, Y.Y., A.G., E.T., D., A.H., A.N., P.S., G.O., A.F., D.R.S., R.S., N.B., J.N.T. and Y.R.; Data curation, Y.Y., A.H., A.N., P.S., J.N.T. and Y.R.; Writing original draft, Y.Y.; Writing and editing, D., Y.R., E.T., G.O., A.G., A.H., A.N., P.S. and J.N.T.; Visualization, Y.Y. and E.T.; Supervision, E.T. All authors have read and agreed to the published version of the manuscript.

**Funding:** This work was funded by the NASA Grant Award Number 80NSSC18K0234, NIST Grant Award Number 70NANB19H024, and the National Science Foundation Award Number 1806494.

**Conflicts of Interest:** The authors declare no conflict of interest.

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
