# Peer review of "Determination of Electron Beam Energy in Measuring the Electron-Impact Ionization Cross Section of He-like Fe24+"

_atoms, doi:10.3390/atoms11030044_

Round 1

Reviewer 1 Report

The manuscript is short but rather well written. The idea is to determine experimentally the electron energy inside the EBIT device that is then used for cross-section calculations for the impact ionization of He-like iron. The authors deduced a correction of the electric potential due to space charge effect which lowers the potential for three specific bu close values of the nominal electron beam energy of 6660, 6710 and 6750 eV. An average  neutralization factor of about 60% was deduced. This correction was then supposed to hold for beam energies in the range 9210-18 000 eV and was used to estimate space-charge corrections and hence corrected energies.

This assumption was not explicitly indicated. Therefore there is a doubt about these corrections. What are the arguments in favor of such a hypothesis ? From table 1 one sees that the space-charge correction decreases from 110 eV to 80 eV when the nominal beam energy is varied from 9210 eV to 18 keV. It is interesting to explain this behavior and how accurate is the assumption of keeping the same neutralization factor when the nominale beam energy is multiplied by a factor 2 to 3. A discussion about the improvement of this point or alternative methods will be very interesting.

There are some other detailed comments which may improve the manuscript.

- Page 2. L63-68. It is not obvious why the authors use experiments at these three nominal beam energies lowers that the energy domain 9-18 keV. It can be better explained. Also may be it would be more interesting for the readers if the term "nominal" is explained.

L72. The greek symbol alpha is better that alpha in K-alpha line.

L77. The electron energy is positive. As the electric potential V is positive, I think one should use the absolute value of the electron charge q_e.

- Page 3.

Figure 1 can be split into two figures to increase the size of each panel because it is a bit hard to distinguish the different curves. It would good if the authors identify the major peaks (which lines?). It may be more interesting to move figure 1 and put it just before figure 2.

Page 5. It does not seem to me that a section is necessary for results as it contains only 5 lines.

In table 1, the nominal energy is designed by E_n while in the text it is designed by E.

The conclusion needs more development like the limitations of these calculations and the perspectives of this work.

Author Response

Review Report Reviewer 1 

Comments and Suggestions for Authors: 

The manuscript is short but rather well written. The idea is to determine experimentally the electron energy inside the EBIT device that is then used for cross-section calculations for the impact ionization of He-like iron. The authors deduced a correction of the electric potential due to space charge effect which lowers the potential for three specific bu close values of the nominal electron beam energy of 6660, 6710 and 6750 eV. An average  neutralization factor of about 60% was deduced. This correction was then supposed to hold for beam energies in the range 9210-18 000 eV and was used to estimate space-charge corrections and hence corrected energies. 

This assumption was not explicitly indicated. Therefore there is a doubt about these corrections. What are the arguments in favor of such a hypothesis ? From table 1 one sees that the space-charge correction decreases from 110 eV to 80 eV when the nominal beam energy is varied from 9210 eV to 18 keV. It is interesting to explain this behavior and how accurate is the assumption of keeping the same neutralization factor when the nominale beam energy is multiplied by a factor 2 to 3. A discussion about the improvement of this point or alternative methods will be very interesting. 

Reply to comments: 

The authors would like to thank the reviewer for carefully reading the document and for the insightful comments. We have made changes to the manuscript and hope that we were able to make it more consistent and clearer for the reader.   

The concept of the neutralization factor has been discussed by several prior publications considering the ion cloud and its dynamics inside electron beam based ionizing devices. These for example include the paper on the “Physics of Electron Beam Ion Traps and Sources" by Currell and Fussmann (IEEE Transaction on Plasma Science 33 (2005) 1763), the paper on the "Space Charge Neutralization in the Ionizing Beam of a Mass Spectrometer," by Plumlee (The Review of Scientific Instruments 28 (1957) 830) and the "Direct Imaging of Highly Charged Ions in an Electron Beam Ion Trap" by Porto, Kink, and Gillaspy (Review of Scientific Instruments 71 (2000) 3050). Based on these model calculations and measurements the constant neutralization factor is generally assumed by the EBIT community. The assumption should hold for situations where the trapping parameters (including the electron beam density, the magnetic field, and the drift tube voltages) are kept the same. We realize however the legitimacy of the comment of the referee and in future measurements we plan to check this assumption. For the main message of the current manuscript, the extrapolation to higher energies is not critical, therefore we have removed those parts and the corresponding table to avoid potential confusion. We thank the referee for pointing this out.  

There are some other detailed comments which may improve the manuscript. 

- Page 2. L63-68. It is not obvious why the authors use experiments at these three nominal beam energies lowers that the energy domain 9-18 keV. It can be better explained. Also may be it would be more interesting for the readers if the term "nominal" is explained. 

Reply: 

We appreciate this comment. We have changed the text and removed the references to the electron beam energies where the ionization cross section measurements took place to avoid confusion.  We also made it clear that the current measurements were taken at lower nominal beam energies of 6660 eV, 6710 eV, and 6750 eV where dielectronic resonances produce satellite lines that can be measured. To make the description clearer we have added a paragraph about DR satellites and included the definition of the nominal beam energy.  

“DR is a two-step resonance process where a continuum electron recombines with an ion while a bound atomic electron is excited. The radiative de-excitation of the doubly-excited state produces the spectral footprints of the resonant excitation that can be observed in the emitted spectra. The specific kinetic energy requirement for the captured free electron, E_{1e}+E_{1b} = E_2 (see Figure 1 for the notation), makes DR produced satellite lines useful for probing the energy of the electron beam of the EBIT. It practically provides two features useful for measuring the space-charge correction. 1) high sensitivity to electron beam energy and 2) low uncertainty on input parameters to theoretical calculations. At NIST, the energy resolution of the recently installed transition-edge-sensor (TES) spectrometer allows the observation of the DR spectral features with high x-ray energy resolution that can be matched to theoretical models.” 

“The potential difference between the cathode of the electron gun and the middle drift tube (MDT) in the trap defines the nominal electron beam energy”  

L72. The greek symbol alpha is better that alpha in K-alpha line. 

Reply: 

Thank you. We have changed the text to use the greek symbol alpha 

L77. The electron energy is positive. As the electric potential V is positive, I think one should use the absolute value of the electron charge q_e. 

Reply: 

Thank you for noticing this. We have changed the text to “where qe is the absolute value of the charge of an electron”. 

- Page 3. 

Figure 1 can be split into two figures to increase the size of each panel because it is a bit hard to distinguish the different curves. It would good if the authors identify the major peaks (which lines?). It may be more interesting to move figure 1 and put it just before figure 2. 

Reply: 

Thank you for this comment. We have split it into two figures to increase the size of each panel for better readability.   

Because each prominent peak in the figure represents numerous DR lines underneath it is difficult to make specific labels as doing so would give the impression that we only used the intensity of the strongest lines. Every theoretical line in this region was included in the comparison. 

We also wanted to clarify that: “Spectra were normalized to the strong feature in the spectra at around 6700 eV. Minimizing the least-squares difference between the intensities of the experimental and theoretical spectra provided the best beam energy matches for the experimental spectra.”  

Page 5. It does not seem to me that a section is necessary for results as it contains only 5 lines. 

Reply: 

Thank you for this comment. We have restructured our paper and removed the table as these energies are indeed not related to the current determination of the neutralization factor. We combined the Results section and the Conclusion to keep the focus on the neutralization factor and the beam energies where the experiment was conducted at. 

In table 1, the nominal energy is designed by E_n while in the text it is designed by E. 

Reply: 

We have removed Table 1. Based on your comment above.  

The conclusion needs more development like the limitations of these calculations and the perspectives of this work. 

Reply: 

Thank you for the suggestion. We combined the Results and Conclusions sections based on the changes we have made to the manuscript and included a description of the perspectives.  

Reviewer 2 Report

In EBIT, it is important to accurately determine the energy of the electron beam, but this is very difficult because of the neutralizing effects of the space charge of the electron beam itself and the space charge of the highly charged ions that are produced. It is commendable that the authors took on this challenge and aimed to establish accurate ionization cross section values and methods for their determination. In this work, the actual beam energy for measurements of ionization cross sections were determined by comparing the observed X-ray spectra with theoretically simulated spectra. However, the manuscript does not address the validity of several key assumptions, nor does it address the validity of the calculated values of the neutralization coefficients. The method of space-charge correction in EBIT is well established to some extent except for how much the effect of neutralization is taken into account. The value of this manuscript should be discussed by the validity of some assumptions and calculations and the ionization cross section values obtained. Since the authors state that this paper focuses on methods for accurately determining the electron beam energy, a detailed description of some assumptions and the validity of the simulations is necessary. The contents of this manuscript should be included in the experimental apparatus and methods chapter of the paper reporting ionization cross sections, and if you wish to complete this part of the manuscript as a stand-alone paper, please provide a detailed description of the determination method.

In follows, my questions for the author and comments on items that should be supplemented are described.

Comment 1

In L84, the value of Vo when assuming a Gaussian distribution is given, please provide a simple explanation that leads to a coefficient of 1.08 and the basis on which a Gaussian distribution can be assumed. Citing reference [10] is not sufficient.

Comment 2

After all, which of the V0 values did you adopt?

Comment 3

Gall's tables are cited in L88-L89, but since this is a doctoral dissertation, it takes more effort than usual for the reader to refer to them. I recommend that you re-cite something like the evidence paper. In particular, since this section argues that Gall's correction is inadequate, a comparison with the results of this study, e.g., how different they are, should be described, even if it is later in this manuscript.

Comment 4

L94-96 You say that corrections can be made by comparing theoretical and experimental values, but can you assure me that the theoretical calculation of thresholds is correct?

Comment 5

Please provide justification for applying the collisional-radiative simulations with the NOMAD code and the rationale for applying this model.  L100-110.

Comment 6

In L131-133, the average neutralization factor is given, but please indicate how the error 5% is derived.

Comment 7

As with comment 6, please clarify in the text, if possible, how the error widths related to the value of corrected Eb shown in Table 1 are calculated.

Comment 8

The last part of the conclusion section, L150-151, is incomprehensible to me. The correction methodology used in this study relies on theoretical assumptions and simulation results. Nevertheless, how can the ionization cross sections measured using this methodology be a benchmarking for the theory?

Author Response

Review Report Reviewer 2 

Comments and Suggestions for Authors: 

In EBIT, it is important to accurately determine the energy of the electron beam, but this is very difficult because of the neutralizing effects of the space charge of the electron beam itself and the space charge of the highly charged ions that are produced. It is commendable that the authors took on this challenge and aimed to establish accurate ionization cross section values and methods for their determination. In this work, the actual beam energy for measurements of ionization cross sections were determined by comparing the observed X-ray spectra with theoretically simulated spectra. However, the manuscript does not address the validity of several key assumptions, nor does it address the validity of the calculated values of the neutralization coefficients. The method of space-charge correction in EBIT is well established to some extent except for how much the effect of neutralization is taken into account. The value of this manuscript should be discussed by the validity of some assumptions and calculations and the ionization cross section values obtained. Since the authors state that this paper focuses on methods for accurately determining the electron beam energy, a detailed description of some assumptions and the validity of the simulations is necessary. The contents of this manuscript should be included in the experimental apparatus and methods chapter of the paper reporting ionization cross sections, and if you wish to complete this part of the manuscript as a stand-alone paper, please provide a detailed description of the determination method. 

In follows, my questions for the author and comments on items that should be supplemented are described. 

Reply: 

We appreciate the reviewer’s insights and comments.  

Based on the comments we rewrote parts of the introduction and experimental sections and refocused our paper for the determination of the neutralization factor at the lower beam energies where the DR resonances are present. We plan to assess the possible scaling of the neutralization factor and the uncertainties associated with that separately when we are going to report the ionization cross section results. In commenting on the validity of the assumptions we would like to refer to prior work by other groups working on this problem as we have also noted in our reply to Reviewer 1.  In particular, the concept of the neutralization factor has been discussed by several prior publications considering the ion cloud and its dynamics inside electron beam based ionizing devices. These for example include the paper on the “Physics of Electron Beam Ion Traps and Sources" by Currell and Fussmann (IEEE Transaction on Plasma Science 33 (2005) 1763), the paper on the "Space Charge Neutralization in the Ionizing Beam of a Mass Spectrometer," by Plumlee (The Review of Scientific Instruments 28 (1957) 830) and the "Direct Imaging of Highly Charged Ions in an Electron Beam Ion Trap" by Porto, Kink, and Gillaspy (Review of Scientific Instruments 71 (2000) 3050). Based on these model calculations and measurements the constant neutralization factor is generally assumed by the EBIT community. The assumption should hold for situations where the trapping parameters (including the electron beam density, the magnetic field, and the drift tube voltages) are kept the same. We realize however the legitimacy of the comments of both referees and in future measurements we plan to check this assumption. For the main message of the current manuscript, the extrapolation to higher energies is not critical, therefore we have removed those parts and the corresponding table to avoid potential confusion. 

Comment 1 

In L84, the value of Vo when assuming a Gaussian distribution is given, please provide a simple explanation that leads to a coefficient of 1.08 and the basis on which a Gaussian distribution can be assumed. Citing reference [10] is not sufficient. 

Reply: 

There have been several previous studies to determine the density distribution of the electron beam and it was found that the Gaussian distribution is a good assumption for the density shape of the beam. In order to address the question by the reviewer we have added a paragraph citing some of these studies.  

“The radial density distribution of the electron beam in the EBIT has been measured previously (see e.g. [14,15]), and it is close to Gaussian in nature. An average value, equal to the density of a uniform cylindrical beam of a certain radius can be used to characterize the electron beam density as it is producing a radial potential that is similar to that of a Gaussian beam [10, 16]. In this work, we assumed that the electron beam density distribution is Gaussian and V0 was calculated accordingly.” 

In regard to the 1.08 factor reference [10] has the details, however, the basic idea is that based on Gauss’ law of electrostatics both the uniform cylindrical electron beam distribution and the Gaussian density distribution electron beam provide similar potentials with a 1.08 factor between the two for V0. 

After all, which of the V0 values did you adopt? 

Reply: 

We use the Gaussian distribution assumption for the rest of the paper and made it clear to the reader through the paragraph we added. We appreciate the reviewer’s request for this clarification. 

Comment 3 

Gall's tables are cited in L88-L89, but since this is a doctoral dissertation, it takes more effort than usual for the reader to refer to them. I recommend that you re-cite something like the evidence paper. In particular, since this section argues that Gall's correction is inadequate, a comparison with the results of this study, e.g., how different they are, should be described, even if it is later in this manuscript. 

Reply: 

Thank you for the comment. Gall’s Ph.D. thesis is easily and freely available on the web by searching the reference we have provided. We used the reference as an example in this section for the space-charge correction that can be applied at various nominal beam energies and beam currents, but they are not essential for the current results. We have modified the text accordingly. 

“The thesis of Gall provides examples and tables for the space-charge correction at different beam energies and beam currents in the NIST EBIT [17].” 

 Comment 4 

L94-96 You say that corrections can be made by comparing theoretical and experimental values, but can you assure me that the theoretical calculation of thresholds is correct? 

Reply: 

We appreciate the comment. Our claim that comparing theoretical resonance electron energies of DR satellite features with measured values is based on the general assumption that the FAC calculations used in this study provide theoretical accuracy on the order of 1 eV, much better than the experimental uncertainties involved. This claim has been well established in the atomic structure theory community and has been tested by comparisons with Relativistic Many-Body Perturbation and Multi-Configuration Dirac Fock calculations.  

In order to clarify the method further we have added a section with an explanation of DR satellite transitions. 

“DR is a two-step resonance process where a continuum electron recombines with an ion while a bound atomic electron is excited. The radiative de-excitation of the doubly-excited state produces the spectral footprints of the resonant excitation that can be observed in the emitted spectra. The specific kinetic energy requirement for the captured free electron, E_{1e}+E_{1b} = E_2 (see Figure 1 for the notation), makes DR produced satellite lines useful for probing the energy of the electron beam of the EBIT. It practically provides two features useful for measuring the space-charge correction. 1) high sensitivity to electron beam energy and 2) low uncertainty on input parameters to theoretical calculations. At NIST, the energy resolution of the recently installed transition-edge-sensor (TES) spectrometer allows the observation of the DR spectral features with high x-ray energy resolution that can be matched to theoretical models.” 

Comment 5 

Please provide justification for applying the collisional-radiative simulations with the NOMAD code and the rationale for applying this model.  L100-110. 

Reply: 

The NOMAD collisional radiative code has been thoroughly tested for modelling the radiation generated by modeling the non-Maxwellian plasma of the EBIT. (see, for example DOI: 10.1585/pfr.8.2503024 , DOI:https://doi.org/10.1103/PhysRevA.74.042514, DOI:https://doi.org/10.1103/PhysRevA.80.010501, DOI: 10.1088/1361-6455/ac44e1 and other publications by the NIST EBIT group). 

Comment 6 

In L131-133, the average neutralization factor is given, but please indicate how the error 5% is derived. 

Reply: 

Thank you for the request for clarification. The experimentally obtained space charge correction is 130 eV, with an uncertainty of 10eV. The 5% uncertainty in the neutralization factor was calculated as a simple propagation of this uncertainty. We included a clarification in the paper. 

“Comparison of the space-charge correction from the 6660 eV and 6710 eV nominal energy experimental spectra with Equation 2 allowed us the determination of the average neutralization factor by the ion cloud, defined by Equation 3, to be 60%± 5%. The uncertainty of the value corresponds to the upper limit of the uncertainty of the space-charge correction to the nominal beam energies. The latter was determined by matching the experimental spectra with a theoretical model of the spectral features.” 

Comment 7 

As with comment 6, please clarify in the text, if possible, how the error widths related to the value of corrected Eb shown in Table 1 are calculated. 

Reply: 

Based on the earlier comment of Reviewer 2 and also by comments of Reviewer 1. We have removed Table 1. and combined the Results and Conclusions sections.   

Comment 8 

The last part of the conclusion section, L150-151, is incomprehensible to me. The correction methodology used in this study relies on theoretical assumptions and simulation results. Nevertheless, how can the ionization cross sections measured using this methodology be a benchmarking for the theory? 

Reply: 

Thank you for the comment. We have reworded our Results and Conclusion sections for more clarity.  

“The neutralization factor determined this way relies on the accurate collisional-radiative modeling of the experimental spectra that include DR satellites of He-like spectral lines. These calculations have been tested in many prior experiments in the EUV and x-ray 
spectral regions. Our future efforts are going to focus on the scaling of the neutralization corrected space-charge shift to beam energies where the ionization cross section determinations will take place. Our measurements provide benchmarks for models of the ion cloud distribution in electron beam based devices like the EBIT.” 

Round 2

Reviewer 2 Report

The revised manuscript is easier to understand because it focuses on DR resonance and the determination of the neutralization factor at low energy. In particular, the addition of the explanation of DR was a good idea.
My question about the experimental error was resolved by the explanation added by the author.
I have confirmed that Gall’s Ph.D. thesis is still cited, but that it can be obtained from ProQuest, so my concerns have been addressed.
Based on the above, I find that this manuscript has been sufficiently improved to merit publication in ATOM. I salute you for your sincere response to my questions and comments, which often may not be accurate, and for revising the manuscript to provide a better paper for ATOM readers.